# Enhancing Healthcare for Sarcoma Patients: Lessons from a Diagnostic Pathway Efficiency Analysis

**DOI:** 10.3390/cancers15194892

**Published:** 2023-10-09

**Authors:** Maria Elyes, Philip Heesen, Georg Schelling, Beata Bode-Lesniewska, Gabriela Studer, Bruno Fuchs

**Affiliations:** 1University Teaching Hospital LUKS, Lucerne, Sarcoma Service, University of Lucerne, 6000 Lucerne, Switzerland; 2University Hospital USZ, Sarcoma Servuce, University of Zurich, 8000 Zurich, Switzerland; 3Pathologie Institut Enge, University of Zurich, 8000 Zurich, Switzerland; 4Kantonsspital Winterthur, Sarcoma Service, 8400 Winterthur, Switzerland

**Keywords:** sarcoma, benign bone tumor, benign soft-tissue tumor, total interval of diagnostic pathway, diagnostic interval, referral patterns, healthcare system, quality management system, MDT/SB-SSN, multidisciplinary Team/Sarcoma Board of the Swiss Sarcoma Network, RWTD/E, real-world-time data evidence

## Abstract

**Simple Summary:**

The total interval of the diagnostic pathway, which consists of the patient interval and the diagnostic interval, describes the time between the first symptom and the final diagnosis. Thus, it could be used as an efficiency marker of a healthcare system. The efficiency of the most expensive health care system in Europe, Switzerland, for bone and soft tissue sarcomas, as well as their benign representatives, has not yet been described. Sarcomas are rare and have a worse outcome than more common tumors. It is assumed that a short total interval leads to a better outcome. Finding out where to start in the total interval to achieve the greatest potential for optimization and to elicit healthcare efficiency is the goal of this study. We have done this by dividing the total interval into its components and looking at their length, as well as potential influencing factors. This revealed that the patient and secondary care interval represent bottlenecks with age, grade, localization, and size being influencing factors of the length of intervals and probability of sarcoma.

**Abstract:**

Sarcomas, rare and with lower survival rates than common tumors, offer insights into healthcare efficiency via the analysis of the total interval of the diagnostic pathway, combining the patient interval (time between the first symptom and visit with a physician) and diagnostic interval (time between first physician visit and histological diagnosis). Switzerland’s healthcare system, Europe’s costliest, lacks research on treating rare conditions, like mesenchymal tumors. This study examines the total interval of the diagnostic pathway for optimization strategies. Analyzing a dataset of 1028 patients presented from 2018 to 2021 to the Swiss Sarcoma Board (MDT/SB-SSN), this retrospective analysis delves into bone sarcoma (BS), soft-tissue sarcoma (STS), and their benign counterparts. Demographic and treatment data were extracted from medical records. The patient interval accounted for the largest proportion of the total interval and secondary care interval for the largest proportion of the diagnostic interval. Age, grade, and localization could be elicited as influencing factors of the length of different components of the total interval. An increasing age and tumor size, as well as the axial localization, could be elicited as factors increasing the probability of sarcoma. The patient and secondary care interval (SCI) offer the greatest potential for optimization, with SCI being the bottleneck of the diagnostic interval. New organizational structures for care work-ups are needed, such as integrated practice units (IPU) as integral part of value-based healthcare (VBHC).

## 1. Introduction

Several studies have examined the diagnostic interval of sarcomas, yet none have specifically characterized this interval within Switzerland, the European country with the highest healthcare costs [1]. This research gap underscores a critical need to comprehensively understand the diagnostic pathway for sarcoma patients in a healthcare system characterized by high costs and a unique geographical and cultural landscape. 

Sarcomas are among the rare diseases with an incidence of 4.43 per 100,000 person-years for soft-tissue sarcoma (STS) and 0.91 per 100,000 person-years for bone sarcoma (BS) [2,3]. Apart from their mesenchymal origin, sarcomas exhibit remarkable heterogeneity, with more than 80 histological subgroups [4] and diverse ages at disease onset, sites of manifestation, and tumor progression aggressiveness. This complexity combined with the rather limited research on rare cancers leads to an incomplete understanding of sarcoma biology, diagnostic challenges, and less effective therapies and guidelines. Consequently, these factors contribute to the observed lower survival rates compared to more common cancer entities [5].

This situation underscores a relevant concern, as the lack of comprehensive insights presents a significant obstacle for implementing both national and international measures. To comprehend potential variations in the treatment of sarcoma patients within the Swiss healthcare framework, it becomes essential to scrutinize the structure of the healthcare system itself. Switzerland’s healthcare system is upheld through a blend of public and private funding. Access to healthcare services necessitates mandatory health insurance for citizens, who are also liable for a significant portion of the healthcare expenses. The Swiss healthcare system is characterized by a high quality of care, great patient satisfaction, extensive patient autonomy in choosing medical service providers, and a wide range of medical service providers [1]. Consequently, patients do not necessarily have to seek a primary care physician first, but can go directly to a secondary care specialist, depending on their insurance model and preferences. The referral patterns of sarcoma patients in Switzerland remain uncharted. Hence, it remains uncertain if the substantial healthcare expenses also lead to a positive outcome in the shape of short diagnostic intervals for uncommon conditions, like mesenchymal tumors. This matter holds significance not just for patients but also for governmental bodies, with potential cost-saving implications.

The diagnostic interval (the time between the first physician visit and a histologically confirmed diagnosis) together with the patient interval (the time between the date of the first symptom and first consultation with a physician) collectively compose the total interval of the diagnostic pathway (the time from the first mesenchymal tumor-related symptom to the histological confirmation of the diagnosis) [6,7,8]. The diagnostic interval describes the referrals from primary care via secondary care to the tertiary care sector. Tertiary care involves specialized medical facilities, such as sarcoma centers for mesenchymal tumors. To counteract the complex nature of mesenchymal tumors, which leads to diagnostic challenges and suboptimal treatment courses and outcomes, the centralization or regionalization of diagnosis and treatment of sarcoma patients is advocated [9,10,11,12]. However, the feasibility of such centralization or regionalization depends on the availability of the necessary logistical capacity, including the presence of sarcoma specialists. Otherwise, there could be a backlog of patients in the tertiary care sector if referrals from the secondary care sector exceed its capacity. 

Early diagnosis is essential for the patient outcome in many cancer entities [13]. This is also true for sarcoma patients, for whom early diagnosis has a positive impact on survival [14]. To ensure a timely diagnosis, it is crucial for the total interval of the diagnostic pathway to be minimized.

To optimally shorten the total interval of the diagnostic pathway, it is imperative to gain a comprehensive understanding of how primary, secondary, and tertiary care intervals are interrelated, what the referral structures are, and therefore, what type of physicians (hospital-based vs. practice-based) are involved in the diagnostic pathway for the diagnosis of a mesenchymal tumor. This aspect remains unexplored to date.

To address which components of the total interval of the diagnostic pathway could be improved and to identify the appropriate targets for optimization efforts, aiming to minimize the duration between the initial symptom onset and diagnosis for patients with mesenchymal tumors, as well as to determine which patients are more likely to have a malignant mesenchymal tumor, this study investigates the various components of the total interval of the diagnostic pathway. These include an evaluation of their length, potential factors influencing the length of intervals, as well as the likelihood of a diagnoses, and an analysis of the involvement of different physicians (hospital-based vs. practice-based).

## 2. Materials and Methods

### 2.1. Study Design

This study represents a retrospective analysis of a prospectively collected dataset (based on a prospectively collected, real-world-time datawarehouse/-lake; Sarconnector^®^ (PH&BF, Zurich, Switzerland) that included bone sarcoma (BS) and soft-tissue sarcoma (STS) patients, as well as patients diagnosed with a benign bone tumor or benign soft-tissue tumor, at a sarcoma center (MDT/SB-SSN) with its associated network, including seven secondary and tertiary care medical institutions in Switzerland, which constitutes the Swiss Sarcoma Network (SSN).

### 2.2. Study Objective

The main objective of this study was to analyze the diagnostic pathway from the first symptom to the histologically confirmed diagnosis in terms of physicians involved, length of the total interval, patient interval, and diagnostic interval, consisting of primary, secondary, and tertiary care intervals, as well as possible influencing factors, such as age, gender, grade, and tumor localization, for the four subgroups, BS, STS, benign bone tumors, and benign soft-tissue tumors. The aim was to use these analyses to describe in which part of the total interval of the diagnostic pathway and for which patients the greatest potential for optimization exists.

### 2.3. Selection Criteria

All consecutive patients presented at the weekly MDT/SB-SSN with a diagnosis of STS, BS, a benign soft-tissue tumor, or a benign bone tumor from 1 January 2018, to 31 December 2021, were included in this study. The diagnoses, which were based on the WHO classification, were divided into benign and malignant, with intermediate tumors categorized as malignant. 

Patients were excluded if records were incomplete. Records were considered incomplete if, for example, no conclusion could be drawn from the available medical records as to the date of the primary and secondary care physician visit (see Figure 1). Since in the Swiss healthcare system, a visit to a primary care physician is not obligatory in every case before a visit to a specialist, patients whose data regarding the primary care interval were not complete were included. This was done because it was not possible to distinguish between (1) the absence of physician-directed care and (2) no documentation of a physician visit in the primary care interval. Named patients were listed as not available (NA) in Figures 3 and 4 under the primary care interval. The same reasoning was used for missing data based on the secondary care interval. These patients were also listed as NA in Figures 3 and 4 under secondary care interval. If it was clear from the medical records that a primary or secondary care physician was not involved (e.g., because it was an incidental finding in the context of other examinations in the secondary care interval or because the referral letter from the general practitioner described it as such), the patients were listed in Figures 3 and 4 under the “Absence of physician-directed care”.

### 2.4. Data Collection

Through a RWTD/E warehouse (Adjumed, Zurich, Switzerland) where the demographic and treatment-specific information of the patients from seven Swiss medical institutions are being collected, 1028 patients were identified. Data on age, sex, the WHO diagnosis, and anatomic region were also obtained from this warehouse. Information on the date of the first symptom that could be attributed to the benign or malignant mesenchymal tumor, the date of the first physician visit, the date of referral from primary to secondary care, the first visit to a secondary care physician, the referral to the sarcoma center in the tertiary care interval, and the date of a histologically confirmed diagnosis were extracted from the medical records. In addition, the medical records were used to determine whether the physician was a practice-based or hospital-based physician in primary and secondary care. Primary care physicians included general practitioners, gynecologists, ophthalmologists, pediatricians, and emergency room physicians. Secondary care physicians included all physicians who were not general practitioners. In PCI and SCI, both practice-based and hospital-based (e.g., physicians in an emergency department) physicians were included. The endpoint in the tertiary care interval was the sarcoma center, which was hospital-based in all cases in the included study population. 

### 2.5. Definition of the Intervals

The definitions of the intervals were adopted from Soomers et al. [6] who adapted the standardized definition proposed by Weller et al. [7] and Olesen et al. [8]. The patient interval (PI) was defined as the time between the first noticed mesenchymal tumor-related symptom and first consultation with a medical doctor. The primary care interval (PCI) was defined as the time between the first physician visit and first secondary referral to a physician of the secondary care. Physicians were divided into practice-based and hospital-based. The secondary care interval (SCI) was defined as the time between the first secondary referral and referral to a specialist sarcoma center. Physicians were divided into practice-based and hospital-based. The timespan from referral to a specialist sarcoma center and the date of the histological diagnosis was defined as the tertiary care interval (TCI). Since the diagnosis of a benign or malignant bone or soft-tissue tumor can also take place outside a sarcoma center, the TCI values were sometimes negative. PCI, SCI, and TCI were summarized as the diagnostic interval (DI). The PI and DI resulted in the total interval of the diagnostic pathway (TIDP) (see Figure 2).

### 2.6. Statistical Analysis

Continuous variables are presented as the median (1st quartile, 3rd quartile), while categorical variables are presented as a number (percentage). Due to the low number of missing data, no missing data imputation was performed. To study the association between clinical variables (age, gender, histological grade, tumor localization, and size) and a bone sarcoma versus soft-tissue sarcoma diagnosis or a benign versus sarcoma diagnosis, logistic regression models were created. To assess the association between clinical variables and the described intervals, linear regression was employed. The normal distribution of variables was assessed visually using histograms or QQ-plots. When continuous data were normally distributed, a t-test was performed, while a Mann–Whitney-U test was performed for non-normally distributed data. Differences between categorical variables were tested using a Chi-square test or using Fisher’s exact test (if the expected value was below 5). A *p*-value < 0.05 was considered statistically significant. All analyses were conducted using R (version 4.3.1).

## 3. Results

### 3.1. Diagnosis Probability Based on Patient and Tumor Traits (See Table 1)

Of the factors studied, age, localization, and size influenced the likelihood of bone sarcoma (BS) versus soft-tissue sarcoma (STS) and the likelihood of a benign versus malignant mesenchymal tumor. Most of the included patients (n = 356) were diagnosed with STS, especially deep STS (n = 296). The median age of the studied population was 56.0 years. With a 1-year increase in age, the likelihood of an STS compared with a BS increased by 3%, which was represented by the lower median age of patients with BS (44.0 years) and benign bone tumors (34.0 years). Similarly, the probability of a diagnosis of a malignant compared with a benign bone or soft-tissue tumor increased by 2% with a 1-year increase in age. The overall gender distribution was balanced (48.9% female), with more male patients (63.4%) having BS. However, gender did not affect the likelihood of being diagnosed with BS compared with STS or of being diagnosed with a malignant compared with a benign mesenchymal tumor. Among the sarcomas, grade G3 was the most common. Tumors were more frequently appendicular in location, although the distribution was more balanced in STS. An axial location increased the likelihood of an STS compared with a BS and of a malignant bone or soft-tissue tumor compared with a benign one. Malignant tumors tended to be larger than benign ones. The larger a tumor was, the more likely it was to be diagnosed as STS. The likelihood of a sarcoma compared with a benign tumor also increased with an increasing tumor size. In most subgroups, the number of cases decreased with an increasing tumor size. 

### 3.2. Patient Interval (PI)

#### 3.2.1. Length (See Table 2)

The patient interval (median, overall 90.0 weeks) was longer than the diagnostic interval (median, overall 46.0 weeks) in all subgroups. The patient interval was significantly shorter for deep STS (median, 8.3 weeks) than for superficial STS (median, 20.7 weeks) (*p* = 0.01). No such difference was observed between BS and STS. No differences in PI length were detected between benign bone and soft-tissue tumors or between superficial and deep soft-tissue tumors. 

#### 3.2.2. Influencing Parameters (See Table 3)

Of the potential influencing parameters investigated, age and localization, each showed a significant effect on the PI length in the benign bone and soft-tissue tumor subgroup, as well as in the overall population. An increasing age correlated significantly with a longer PI in the overall population and in soft-tissue tumors (*p* = 0.047 and *p* = 0.04, respectively). The PI was longer in benign bone tumors for an axial localization rather than for a appendicular localization (*p* = 0.002).

### 3.3. Diagnostic Interval (DI), Primary Care Interval (PCI)

#### 3.3.1. Length (See Table 2)

The primary care interval was the shortest of the diagnostic intervals in all subgroups (median, overall 4.0 weeks). The subgroups of sarcomas showed comparable lengths of PCI, with BS (median, 0.6 weeks) having the median longest and superficial STS (median, 0.0 weeks) have the shortest, without statistical significance. Benign mesenchymal tumors also showed comparable lengths of the PCI, with bone tumors (median, 0.8 weeks) having the longest median and superficial soft-tissue tumors (median, 0.3 weeks) having the shortest, again without statistical significance. PCIs from benign mesenchymal tumors were slightly longer on average than comparable subsets of malignant mesenchymal tumors.

#### 3.3.2. Influencing Parameters (See Table 3)

Of the potential influencing parameters investigated, only localization in the STS subgroup showed a significant effect on the PCI length. The axial tumor localization showed a significantly shorter PCI for STS compared to an appendicular localization (*p* = 0.03).

### 3.4. Diagnostic Interval (DI), Secondary Care Interval (SCI)

#### 3.4.1. Length (See Table 2)

The secondary care interval accounted for the largest proportion of the diagnostic interval for sarcomas (median, overall 26.0 weeks). BS (median, 2.2 weeks) had significantly shorter SCIs than STS (median, 4.3 weeks) (*p* = 0.005); again, the SCI of deep STS (median, 3.9 weeks) was significantly shorter than that of superficial STS (median, 8.1 weeks) (*p* = 0.01). Among benign mesenchymal tumors, SCI represented the largest proportion of the DI for the benign soft-tissue tumor group (although this is likely due to deep soft-tissue tumors). For superficial soft-tissue tumors, the lengths of the SCI and TCI were comparable.

#### 3.4.2. Influencing Parameters (See Table 3)

None of the potential influencing parameters investigated had a significant influence on the SCI of any subgroup.

### 3.5. Diagnostic Interval (DI), Tertiary Care Interval (TCI)

#### 3.5.1. Length (See Table 2)

Sarcomas showed significant differences between BS and STS and between deep and superficial STS in the length of the TCI. BS (median, 2.1 weeks) showed a significantly longer TCI than STS (median, 1.3 weeks) (*p* = 0.006). In STS, in turn, the TCI was significantly shorter for superficial STS (median, 0.9 weeks) than for deep STS (median, 1.6 weeks) (*p* < 0.01). Such differences in TCI were not observed for benign bone and soft-tissue tumors. TCIs of malignant mesenchymal tumors were slightly shorter on average than those of the comparable subset of benign tumors.

#### 3.5.2. Influencing Parameters (See Table 3)

Of the potential influencing parameters investigated, grade and localization had a significant effect on the TCI length in the overall population and the STS subgroup. In the overall population, high-grade tumors had a significantly longer TCI (*p* = 0.02). Axial tumor localization showed a significantly shorter TCI for STS compared to an appendicular localization (*p* = 0.03). This was also reflected in the TCI of deep STS (*p* = 0.03).

### 3.6. Total Interval (TI)

#### 3.6.1. Length (See Table 2)

Total intervals were shorter for sarcomas than for benign tumors. The shortest TI was observed for both malignant and benign tumors of deep soft-tissue tumors (median, 20.9 and 43.0 weeks, respectively). The same was true for superficial soft-tissue tumors, which had the longest TI of both malignant and benign tumors (median, 34.8 and 138.1 weeks, respectively). However, no significant differences in the length of the TI were observed between the subgroups of benign tumors and between the subgroups of malignant tumors.

#### 3.6.2. Influencing Factors (See Table 3)

Of the potential influencing factors investigated, only age had a significant effect on the length of the TI, but this was only true for benign soft-tissue tumors and benign deep soft-tissue tumors. An increasing age correlated significantly with a longer TI for benign soft-tissue tumors and benign deep soft-tissue tumors (*p* = 0.004 and *p* = 0.004, respectively). 

### 3.7. Involved Physicians in the Primary Care Interval (PCI) (See Figure 3 and Figure 4)

The PCI showed differences between benign and malignant mesenchymal tumors with respect to the involvement of physicians, as well as their localization in hospitals and medical practices. For malignant tumors (87.50% to 95.12%), PCI physicians were involved more frequently on average than for benign tumors (79.65 to 87.80%). In this regard, PCI physicians were visited more often for BS (95.12%) and superficial STS (95.00%) than for deep STS (87.50%). Benign mesenchymal tumors showed a similar pattern. PCI physicians were seen most often for benign superficial soft-tissue tumors (87.80%) and benign bone tumors (86.89%) and slightly less often for benign deep soft-tissue tumors (79.65%). In the PCI, the physicians consulted were more often practice-based. For malignant mesenchymal tumors (9.65 to 26.92%), physicians were more often hospital-based relative to benign mesenchymal tumors (0.00 to 9.43%).

### 3.8. Involved Physicians in the Secondary Care Interval (SCI) (See Figure 3 and Figure 4)

The SCI showed differences between benign and malignant mesenchymal tumors with respect to the involvement of physicians, as well as their localization in hospitals and medical practices. For malignant tumors (71.28 to 81.67%), SCI physicians were involved more frequently on average than for benign tumors (62.30 to 63.41%). Here, SCI physicians were more frequently involved in superficial STS (81.67%) than in BS (71.95%) or deep STS (71.28%). Benign mesenchymal tumors showed a similar pattern. SCI physicians were most frequently consulted for benign superficial soft-tissue tumors (63.41%), followed by benign deep soft-tissue tumors (62.79%) and benign bone tumors (62.30%). In SCI, physicians consulted were more often hospital-based than in the PCI. In malignant mesenchymal tumors (73.47 to 83.41%), physicians were more often hospital-based relative to benign mesenchymal tumors (68.42 to 76.92%).

## 4. Discussion

This study addresses a significant gap in the literature by comprehensively analyzing the total interval for mesenchymal tumors in patients, particularly focusing on BS and STS, as well as their benign counterparts. The total interval of the diagnostic pathway, a complex measure influenced by diverse tumor-, patient-, and management-specific factors, has been dissected into its components. Notably, this study is the first to explore total intervals for benign mesenchymal tumors. 

The patient interval emerges as a key determinant of the total interval of the entire diagnostic pathway, consistently occupying a major share across subgroups. Among malignant tumors, the secondary care interval assumes prominence in the diagnostic interval. Age, grade, and localization were identified as factors influencing the interval durations of only some intervals, demonstrating the heterogeneity of mesenchymal tumors. A novel finding is the higher involvement of hospital-based practitioners in the diagnosis of sarcomas compared to benign mesenchymal tumors, possibly due to the more severe and urgent symptoms they exhibit that result in more frequent visits to the emergency ward. These insights contribute to a better understanding of the total interval in mesenchymal tumors.

The existing literature shows a wide range in the length of the different intervals of the diagnostic pathway of malignant mesenchymal tumors. The median values (including BS and STS) of the intervals in the present study are in the lower-to-middle range of values in the literature [15,16,17,18,19,20,21,22,23,24,25,26,27,28,29,30,31,32,33,34,35,36,37,38,39,40], indicating a rather efficient healthcare system, which, however, still has potential for optimization. The diagnostic pathway has not been previously analyzed for benign mesenchymal tumors; thus, no comparative values are available. As in the SURVSARC Study [41], the patient interval accounted for the largest proportion of the total interval in this study. This could also be observed in other cancer entities in the literature [42]. In particular, the patient interval accounted for a large proportion of the total interval for benign tumors, which is very important because the greatest potential for optimization lies in shortening the length of the patient interval. Important factors influencing the patient interval were patient age and tumor localization. Higher age has already been seen to be associated with longer intervals in some studies [16,19,27,29,30,43], although there are also studies that found no association [44] or even an opposite association [41]. In our study, there were 38 pediatric tumors (patient age, 2–18 years), 31 bone tumors (20 BSs, 11 benign bone tumors), 7 soft-tissue tumors (2 STSs; one superficial and one deep; 5 benign soft-tissue tumors, all with a deep location). The numbers were too low to compare pediatric with adult tumors.

In the diagnostic interval, the secondary care interval represented the largest proportion in terms of time. Already, Smolle et al. could show that examinations outside a sarcoma center led to a delay [45]. The visit to a GP compared to an emergency ward was associated with a longer primary care interval in the study by Goyal et al. [16]. The present study cannot confirm this; on the contrary, the primary care interval in which practice-based physicians were most frequently visited turned out to be particularly short, reflecting that GPs in Switzerland refer patients for bone and soft-tissue tumors in the shortest possible time. This is a very important finding: in the diagnostic interval, specialists outside a sarcoma center generate the bottleneck rather than primary care physicians. Axial localization leads to shorter patient and diagnostic intervals. Considering that the CNS is also axially located, symptoms are therefore already noticeable with small tumor masses, little room is left for surgery, and assuming that the treating physicians are aware of this, it becomes clear that faster action is required with an axially located tumor.

A longer total interval of the diagnostic pathway has been associated with lower survival by Bandyopadhyay et al. and Ferrari et al. [46,47]. In their study of primary pulmonary artery sarcoma, which had a median total interval of 14.3 weeks, Bandyopadhyay et al. showed a 46% increase in the odds of death when the length of the total interval was doubled [46]. Ferrari et al., in their study on STS in children and adolescents who had a median total interval of 8 weeks, showed a significant negative impact on survival with an increasing length of the total interval (*p* = 0.002) [47]. Translating this for the current examined cohort, where the median length of the total interval was 22.8 weeks for BS and 23.3 weeks for STS, it can be assumed that the survival rate could be increased by shortening the total interval of the diagnostic pathway. However, it is essential to note that a comprehensive investigation of this effect would be necessary. Moreover, aside from its direct impact on survival, a shorter total interval is also desirable due to its influence on patient well-being in cases where the diagnosis remains uncertain [48]. However, it must be taken into account that the patient interval accounts for a larger part of the total interval of the diagnostic pathway than the diagnostic interval.

The patient interval, which had a median duration of 90.0 weeks for this study’s overall population, was nearly twice as long as the diagnostic interval, which had a median duration of 46.0 weeks and represented the largest delay in the total interval of the diagnostic pathway. This underscores that the primary issue does not lie with individual physicians, but rather with the referral process and therefore the structure of the healthcare system itself. The high investments in the healthcare system appear to be insufficient in promptly identifying sarcoma patients. The repeatedly mentioned complexity of the work-up and treatment of patients with sarcoma is greatly explained by the fact that sarcomas do not form a conventional medical discipline per se. Addressing the needs of sarcoma patients necessitates a comprehensive approach via a multidisciplinary team (MDT), and a physician head coordinating among disciplines is crucial. Integrated practice units (IPUs) could provide a solution. Here, the focus is on a problem rather than a discipline [49,50]. By bringing together different health professionals in a unified structured organization, challenges, like cumbersome referrals, could be surmounted, potentially leading to shorter intervals. Further, the patient interval could also be shortened by health professionals helping patients to recognize problems as such [51]. Subsequently, the diagnostic interval would also be shortened, as there would be no need for cumbersome referrals. Thus, the secondary care interval, which is the largest part of the diagnostic interval, could be optimized. In terms of value-based healthcare (VBHC), health outcomes, such as quality of life, could account for a larger share of costs by intercepting patients before they are plagued by unpleasant symptoms for a long time [52]. In addition, costs for nontargeted investigations could be saved. Therefore, reorganizing healthcare structures according to the VBHC principles may greatly enhance the work-up of sarcoma patients [51]. 

This study reflects the contact of patients with a sarcoma center; therefore, the numbers per subgroup are not balanced, which is a limitation of this study. For example, STS is many times more frequent than BS. This could have the consequence that effects of the investigated potential influencing factors did not show up in the smaller subgroups, although they would be present. In addition, a selection bias was found for those patients who presented at a sarcoma center. That is, someone thought of the possibility of a sarcoma diagnosis during the diagnostic interval and involved a sarcoma center. Patients for whom this possibility was not considered may never have been diagnosed with a mesenchymal tumor, thereby remaining within the diagnostic interval indefinitely.

Further investigation is needed to determine the reasons for the delays in the patient and secondary care interval. Regarding the patient interval, the perceived symptoms could be investigated, as well as the reasons that led to the consultation with a physician. Concerning the secondary care interval, a breakdown of the physicians visited in the primary and secondary care interval regarding their specialization, as well as the examinations performed, would be interesting to determine, on the one hand, to whom optimization approaches should be directed and, on the other hand, to determine the correlation of examinations performed with the length of the intervals. In this way, it would be possible to determine which investigations are appropriate and which could be dispensed with, thus saving costs. In addition, the correlation between patient outcome in this study population could be analyzed to confirm or reject the literary data correlating the outcome with the length of the total interval of the diagnostic pathway.

## 5. Conclusions

In Switzerland’s efficient healthcare system, cost does not guarantee an expedited sarcoma diagnosis, possibly due to its multidisciplinary nature. Key factors, such as an older age, larger tumor size, and axial localization are associated with a higher malignancy risk, underscoring the need for shorter diagnostic intervals. Further research is essential for guiding clinicians with sarcoma suspicions. To improve patient outcomes through reduced total and diagnostic intervals, focus must be placed on shortening the patient and secondary care intervals. This necessitates targeted patient education and specialized physician training. In light of our findings, we advocate for the regionalization or centralization of sarcoma care. While secondary care institutions need not be categorically excluded from sarcoma management, their involvement should be contingent upon active collaborations with a multidisciplinary team or sarcoma board from a tertiary care institution, particularly when complex treatments are required. Given these considerations, the logical next advancement for a sarcoma center is the establishment of Integrated Practice Units (IPUs), in alignment with Value-Based Health Care (VBHC) principles. IPUs offer the added benefits of transparently assessing and sharing treatment metrics and quality indicators within a collaborative network. 

## Figures and Tables

**Figure 1 cancers-15-04892-f001:**
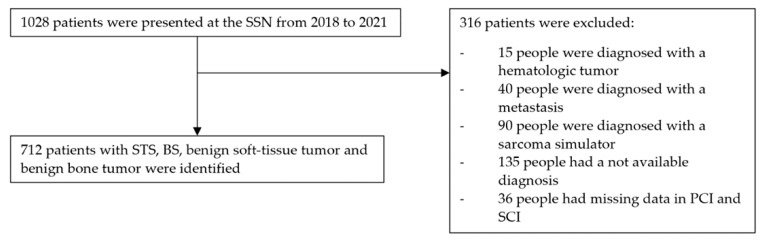
Flow chart of the patient inclusion progress.

**Figure 2 cancers-15-04892-f002:**
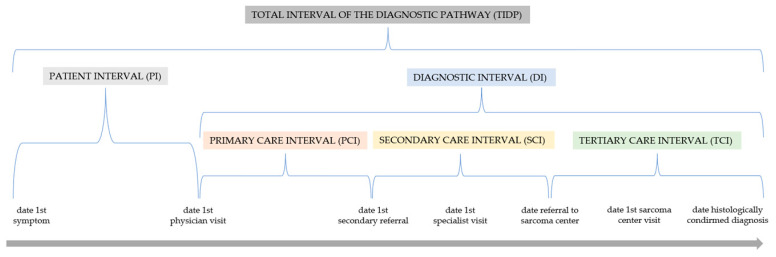
Time intervals from first symptom to the visit to a sarcoma center. Adopted from Soomers et al. 2020 [6]. Patient interval: time between date of first symptom and first visit to a physician. Primary care interval: time between first physician visit and first secondary referral to a specialized physician. Secondary care interval: time between first secondary referral and referral to a specialist sarcoma center. Tertiary care interval: time between referral to a specialist sarcoma center and the date of histological diagnosis. Diagnostic interval: time between first physician visit and histological diagnosis. Total interval of the diagnostic pathway (TIDP): time from first symptom to histological diagnosis.

**Figure 3 cancers-15-04892-f003:**
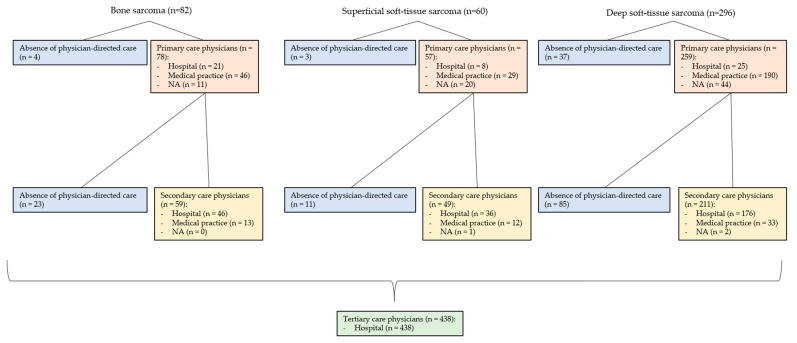
Referral pattern of bone sarcoma, superficial soft-tissue sarcoma, and deep soft-tissue sarcoma.

**Figure 4 cancers-15-04892-f004:**
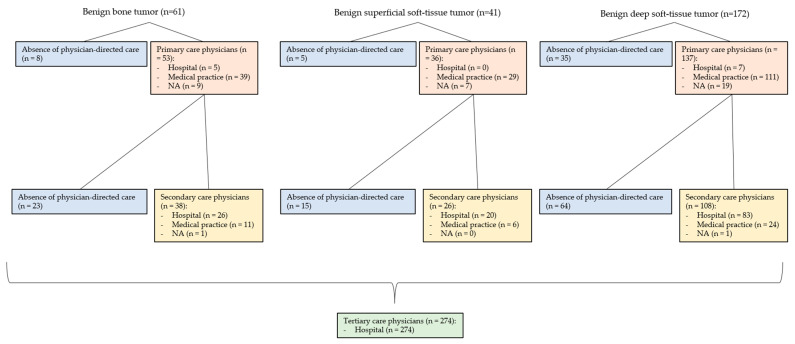
Referral pattern of benign bone tumors, benign superficial soft-tissue tumors, and benign deep soft-tissue tumors.

**Table 1 cancers-15-04892-t001:** Diagnosis probability based on patient and tumor traits.

	OVERALL	BONE SARCOMA	LIKELIHOOD OF BONE SARCOMA VS. SOFT- TISSUE SARCOMA ^a^			SOFT-TISSUE SARCOMA			LIKELIHOOD OF SARCOMA VS. BENIGN TUMOR ^b^			BENIGN BONE TUMOR	BENIGN SOFT-TISSUE TUMOR		
			OR	95% CI	*p*-Value	Deep and Superficial	Deep	Superficial	OR	95% CI	*p*-Value		Deep and Superficial	Deep	Superficial
	n = 712	n = 82				n = 356	n = 296	n = 60				n = 61	n = 213	n = 172	n = 41
**Age, years**	56.0 (40.0, 68.0)	44.0 (19.0, 65.0)	1.03	1.02, 1.04	**<0.001**	60.0 (46.0, 72.0)	60.0 (46.0, 72.0)	61.5 (42.3, 74.3)	1.02	1.01, 1.02	**<0.001**	34.0 (23.0, 45.0)	55.0 (44.0, 63.0)	56.0 (44.0, 65.0)	54.0 (44.0, 61.0)
**Female, (%)**	348 (48.9%)	30 (36.6%)	1.09	0.82, 1.47	0.5	178 (50.0%)	145 (49.0%)	33 (55.0%)	0.87	0.64, 1.17	0.3	30 (49.2%)	110 (51.6%)	90 (52.3%)	20 (48.8%)
**Grade**												not applicable	not applicable	not applicable	not applicable
** *G1, (%)* **	74 (16.9%)	7 (8.5%)				67 (18.8%)	52 (17.6%)	15 (25.0%)							
** *G2, (%)* **	54 (12.3%)	5 (6.1%)	1.17	0.37, 4.07	0.8	49 (13.8%)	41 (13.9%)	8 (13.3%)							
** *G3, (%)* **	126 (28.8%)	26 (31.7%)	0.46	0.18, 1.03	0.07	100 (28.1%)	85 (28.7%)	15 (25.0%)							
** *NA* **	184 (42.0%)	44 (53.7%)				140 (39.3%)	118 (39.8%)	22 (36.7%)							
**Region**			2.91	2.11, 4.04	**<0.001**				2.34	1.67, 3.30	**<0.001**				
** *appendicular* **	469 (65.9%)	65 (79.3%)				193 (54.2%)	165 (55.7%)	28 (46.7%)				50 (82.0%)	161 (75.6%)	131 (76.2%)	30 (73.2%)
** *axial* **	243 (34.1%)	17 (20.7%)				163 (45.8%)	131 (44.3%)	32 (53.3%)				11 (18.0%)	52 (24.4%)	41 (23.8%)	11 (26.8%)
**Size, mm**	60.0 (34.3, 102.0)	60.0 (39.5, 85.0)	1.01	1.00, 1.01	**<0.001**	70.0 (32.0, 124.0)	86.0 (45.0, 130.0)	28.0 (20.0, 44.0)	1.00	1.00, 1.01	**0.001**	31.5 (11.5, 50.5)	60.0 (38.3, 97.3)	61.0 (39.0, 100.5)	54.0 (35.5, 79.5)
** *0–50 mm, n* **	247 (34.7%)	25 (30.5%)				106 (29.8%)	70 (23.6%)	36 (60.0%)				33 (54.1%)	83 (39.0%)	65 (37.8%)	18 (43.9%)
** *51–100 mm, n* **	179 (25.1%)	30 (36.6%)				67 (18.8%)	59 (19.9%)	8 (13.3%)				10 (16.4%)	72 (33.8%)	55 (32.0%)	17 (41.5%)
** *101–150 mm, n* **	91 (12.8%)	11 (13.4%)				50 (14.0%)	50 (16.9%)	0 (0%)				1 (1.6%)	29 (13.6%)	27 (15.7%)	2 (4.9%)
** *>150 mm, n* **	57 (8.0%)	1 (1.2%)				42 (11.8%)	41 (13.9%)	1 (1.7%)				0 (0%)	14 (6.6%)	13 (7.5%)	1 (2.4%)
** *NA* **	138 (19.4%)	15 (18.3%)				91 (25.6%)	76 (25.7%)	15 (25.0%)				17 (27.9%)	15 (7.0%)	12 (7.0%)	3 (7.3%)

^a^: The likelihood of STS (deep and superficial) vs. BS was determined. STS represented the reference. ^b^: The likelihood of sarcoma (STS and BS) vs. benign mesenchymal tumors (benign bone and soft-tissue tumors) was determined. Sarcoma represented the reference.

**Table 2 cancers-15-04892-t002:** Length of patient, diagnostic, primary care, secondary care, tertiary care, and total interval in weeks.

	OVERALL	BONE SARCOMA		SOFT-TISSUE SARCOMA				BENIGN BONE TUMOR		BENIGN SOFT-TISSUE TUMOR			
				Deep and Superficial	Deep		Superficial			Deep and Superficial	Deep		Superficial
	n = 712	n = 82	*p*-Value ^c^	n = 356	n = 296	*p*- Value ^d^	n = 60	n = 61	*p*- Value ^e^	n = 213	n = 172	*p*- Value ^f^	n = 41
**Patient Interval, weeks**	90.0 (22.0, 284.0)	7.8 (2.7, 27.5)	0.46	8.8 (2.1, 29.0)	8.3 (2.0, 24.4)	**0.01**	20.7 (4.2, 130.6)	19.1 (4.3, 52.1)	0.17	21.6 (6.4, 109.6)	19.8 (6.3, 75.1)	0.22	29.9 (9.0, 176.4)
**Diagnostic Interval, weeks**	46.0 (25.5, 95.5)	7.6 (3.1, 14.2)	0.89	6.7 (3.7, 13.3)	6.9 (3.9, 13.7)	0.22	5.7 (3.6, 9.3)	19.8 (6.8, 79.7)	**0.005**	6.0 (3.6, 13.4)	6.0 (3.6, 14.6)	0.35	5.6 (3.6, 9.5)
**Primary** **Care** **Interval,** **weeks**	4.0 (0.0, 18.5)	0.6 (0.1, 6.5)	0.14	0.4 (0.0, 1.4)	0.4 (0.0, 1.3)	0.31	0.0 (0.0, 1.4)	0.8 (0.0, 44.9)	0.30	0.7 (0.0, 3.1)	0.7 (0.0, 4.4)	0.15	0.3 (0.0, 1.0)
**Secondary** **Care Interval,** **weeks**	26.0 (12.0, 57.0)	2.2 (0.9, 6.6)	**0.005**	4.3 (2.1, 9.1)	3.9 (1.9, 8.1)	**0.01**	8.1 (4.9, 10.2)	2.6 (1.0, 10.7)	0.47	3.5 (1.6, 7.5)	3.9 (1.7, 10.0)	0.14	2.6 (1.5, 3.8)
**Tertiary Care** **Interval,** **weeks**	14.0 (5.0, 26.3)	2.1 (1.0, 3.7)	**0.006**	1.3 (-0.6, 3.4)	1.6 (-0.2, 3.6)	**0.01**	0.9 (-3.3, 1.9)	3.1 (2.0, 8.1)	0.14	2.6 (1.8, 4.1)	2.6 (1.7, 4.0)	0.36	2.7 (1.9, 5.8)
Total Interval, weeks	213.0 (84.0, 762.2)	22.8 (11.9, 56.7)	0.82	23.3 (10.4, 59.4)	20.9 (10.4, 55.3)	0.07	34.8 (12.3, 148.0)	100.5 (48.1, 206.6)	0.22	48.2 (17.7, 193.3)	43.0 (14.7, 150.6)	**0.04**	138.1 (29.1, 304.4)

^c^: The *p*-value was calculated based on a Wilcoxon rank sum test with a continuity correction between STS (deep and superficial) and BS. ^d^: The *p*-value was calculated based on a Wilcoxon rank sum test with a continuity correction between deep STS and superficial STS. ^e^: The *p*-value was calculated based on a Wilcoxon rank sum test with a continuity correction between benign soft-tissue tumors (deep and superficial) and benign bone tumors. ^f^: The *p*-value was calculated based on a Wilcoxon rank sum test with a continuity correction between deep benign soft-tissue tumors and superficial benign soft-tissue tumors.

**Table 3 cancers-15-04892-t003:** Influence of age, gender, grade, and localization on intervals.

		PI ^g^						DI ^h^						TI ^l^	
					PCI ^i^			SCI ^j^			TCI ^k^				
	Beta	95% CI	*p*-Value	Beta	95% CI	*p*-Value	Beta	95% CI	*p*-Value	Beta	95% CI	*p*-Value	Beta	95% CI	*p*-Value
**Overall (n = 712)**															
**Age**	7.07	0.08, 14.05	**0.047**	−0.91	−3.36, 1.54	0.46	0.71	−2.31, 3.73	0.65	−0.28	−1.77, 1.21	0.71	6.79	−1.03, 14.61	0.09
**Gender** **male**	reference	reference	reference	reference	reference	reference	reference	reference	reference	reference	reference	reference	reference	reference	reference
**female**	86.53	−185.00, 358.05	0.53	−45.41	−142.57, 51.74	0.36	46.20	−68.73, 161.13	0.43	−11.31	−67.84 45.22	0.69	168.90	−132.39, 470.11	0.27
**Grade** **G1**	reference	reference	reference	reference	reference	reference	reference	reference	reference	reference	reference	reference	reference	reference	reference
**G2**	−392.10	−1039.97, 255.67	0.24	−17.17	−258.50, 224.15	0.89	17.17	−261.18, 295.52	0.90	−28.42	−149.59, 92.76	0.65	−498.92	−1140.89, 143.05	0.13
**G3**	−384.70	−911.59, 142.23	0.15	−77.84	−261.66, 105.98	0.41	−18.07	−240.93, 204.78	0.87	116.81	17.25, 216.36	**0.02**	−470.85	−994.55, 52.85	0.08
**Localization** **appendicular**	reference	reference	reference	reference	reference	reference	reference	reference	reference	reference	reference	reference	reference	reference	reference
**axial (head,** **neck, trunk)**	−177.51	−463.87, 108.86	0.224	−46.46	−147.89, 54.97	0.37	−21.42	−141.14, 98.29	0.73	−84.47	−143.12, −25.82	0.41	−264.11	−583.97, 55.75	0.11
**Bone sarcoma (n = 82)**															
**Age**	5.61	−13.44, 24.66	0.56	−1.48	−10.59, 7.63	0.75	−0.14	−0.95, 0.68	0.74	−0.13	−0.39, 0.13	0.31	7.55	−12.71, 27.80	0.46
**Gender** **male**	reference	reference	reference	reference	reference	reference	reference	reference	reference	reference	reference	reference	reference	reference	reference
**female**	88.12	−830.06, 1006.30	0.85	−53.64	−494.73, 387.45	0.81	2.72	−36.48, 41.93	0.89	−8.45	−21.05, 4.14	0.19	−42.57	−1017.19, 932.04	0.93
**Grade** **G1**	reference	reference	reference	reference	reference	reference	reference	reference	reference	reference	reference	reference	reference	reference	reference
**G2**	−37.05	−2366.05, 2291.95	0.98	−107.67	−1167.15, 951.82	0.84	−37.95	−129.28, 53.37	0.41	−12.80	−42.96, 17.36	0.40	−122.40	−2409.75, 2164.95	0.92
**G3**	48.49	−1679.16, 1776.14	0.96	−90.88	−724.04, 542.28	0.77	−21.64	−82.91, 39.62	0.48	−16.05	−38.39, 6.30	0.16	−15.80	−1815.48, 1783.88	0.99
**Localization** **appendicular**	reference	reference	reference	reference	reference	reference	reference	reference	reference	reference	reference	reference	reference	reference	reference
**axial (head,** **neck, trunk)**	−187.90	−1424.36, 1048.56	0.76	472.71	−88.70, 1034.13	0.10	−7.66	−51.97, 36.65	0.73	−2.06	−16.31, 12.19	0.77	−45.29	−1259.59, 1169.00	0.94
**Soft-tissue** **sarcoma (n = 356)**															
**Age**	4.67	−6.17, 15.51	0.40	0.55	−1.51, 2.60	0.60	0.78	−3.91, 5.47	0.74	1.11	−1.71, 3.93	0.44	3.90	−6.60, 14.39	0.47
**Gender** **male**	reference	reference	reference	reference	reference	reference	reference	reference	reference	reference	reference	reference	reference	reference	reference
**female**	−42.41	−427.41, 342.60	0.83	−24.30	−101.26, 52.67	0.53	−22.46	−189.49, 144.57	0.79	−16.93	−115.72, 81.87	0.74	−18.80	−391.39, 353.80	0.92
**Grade G1**	reference	reference	reference	reference	reference	reference	reference	reference	reference	reference	reference	reference	reference	reference	reference
**G2**	−430.19	−1119.79, 259.41	0.22	−5.59	−157.29, 146.12	0.45	24.29	−291.39, 339.96	0.88	−28.36	−192.80, 136.07	0.74	−525.67	−1148.01, 96.67	0.10
**G3**	−420.90	−997.45, 155.64	0.15	−73.77	−194.34, 46.79	0.94	−17.01	−277.62, 243.60	0.90	134.37	−3.76, 272.50	0.06	−498.96	−1015.67, 17.75	0.06
**Localizationappendicular**	reference	reference	reference	reference	reference	reference	reference	reference	reference	reference	reference	reference	reference	reference	reference
**axial (head, neck, trunk)**	−220.20	−604.58, 164.21	0.26	−84.99	−160.21, −9.77	**0.03**	−109.96	−275.68, 55.75	0.19	−112.59	−210.99, −14.18	**0.03**	−297.40	−672.92, 78.11	0.12
**Deep soft-tissue sarcoma (n = 296)**															
**Age**	5.03	−6.50, 16.57	0.39	1.43	−0.97, 3.84	0.24	0.72	−4.95, 6.38	0.80	0.63	−2.83, 4.10	0.72	4.06	−7.91, 16.02	0.51
**Gender** **male**	reference	reference	reference	reference	reference	reference	reference	reference	reference	reference	reference	reference	reference	reference	reference
**female**	−77.25	−479.40, 324.89	0.71	−40.20	−128.01, 47.61	0.37	−18.11	−213.53, 177.31	0.86	−40.66	−158.05, 76.72	0.50	−66.69	−481.55, 348.17	0.75
**Grade** **G1**	reference	reference	reference	reference	reference	reference	reference	reference	reference	reference	reference	reference	reference	reference	reference
**G2**	−222.90	−966.79, 520.90	0.56	−32.83	−218.91, 153.24	0.73	32.40	−354.26, 419.07	0.87	−54.82	−251.40, 141.76	0.58	−384.41	−1088.57, 319.74	0.28
**G3**	−302.30	−928.03, 323.33	0.34	−114.18	−260.68, 32.32	0.13	−21.76	−339.89, 296.37	0.89	141.65	−24.05, 307.35	0.09	−477.53	−1060.61, 105.54	0.11
**Localization** **appendicular**	reference	reference	reference	reference	reference	reference	reference	reference	reference	reference	reference	reference	reference	reference	reference
**axial (head, neck, trunk)**	−292.70	−694.86, 109.40	0.15	−83.18	−169.70, 3.33	0.06	−124.82	−319.22, 69.58	0.21	−128.24	−245.63, −10.84	**0.03**	−314.20	−735.48, 107.05	0.14
**Superficial soft-tissue sarcoma (n = 60)**															
**Age**	9.08	−22.59, 40.75	0.56	−3.20	−6.91, 0.52	0.09	0.13	−1.18, 1.43	0.84	2.71	−0.64, 6.06	0.11	3.61	−18.69, 25.91	0.75
**Gender** **male**	reference	reference	reference	reference	reference	reference	reference	reference	reference	reference	reference	reference	reference	reference	reference
**female**	−244.50	−1550.78, 1061.81	0.71	62.06	−98.57, 222.68	0.43	−21.07	−73.30, 31.16	0.41	87.35	−47.66, 222.35	0.20	109.40	−765.23, 984.12	0.80
**Grade** **G1**	reference	reference	reference	reference	reference	reference	reference	reference	reference	reference	reference	reference	reference	reference	reference
**G2**	−1031.80	−3045.04, 981.47	0.30	36.96	−205.08, 279.01	0.75	−18.82	−95.17, 57.53	0.61	102.78	−128.78, 334.35	0.38	−1004.50	−2403.36, 394.36	0.16
**G3**	−414.60	−2092.34, 1263.09	0.62	−9.45	−224.29, 205.39	0.93	−1.07	−68.84, 66.70	0.97	93.13	−100.01, 286.27	0.34	−352.70	−1545.67, 840.24	0.56
**Localization** **appendicular**	reference	reference	reference	reference	reference	reference	reference	reference	reference	reference	reference	reference	reference	reference	reference
**axial (head,** **neck, trunk)**	−1.20	−1214.74, 1212.34	0.998	−91.68	−243.05, 59.69	0.22	−11.61	−61.93, 38.71	0.64	−52.58	−187.69, 82.53	0.44	−371.90	−1235.52, 491.64	0.39
**Benign bone** **tumor (n = 61)**															
**Age**	12.03	0.31, 23.75	**0.045**	0.3656	−17.12, 17.85	0.97	−12.059	−32.23, 8.11	0.23	−5.180	−16.54, 6.18	0.35	−11.95	−46.97, 23.06	0.48
**Gender** **male**	reference	reference	reference	reference	reference	reference	reference	reference	reference	reference	reference	reference	reference	reference	reference
**Female**	30.04	−300.40, 360.48	0.86	−367.90	−798.01, 62.17	0.09	337.80	−253.09, 928.69	0.25	271.00	−145.74, 687.74	0.19	314.00	−1011.63, 1639.63	0.63
**Grade**	not applicable	not applicable	not applicable	not applicable	not applicable	not applicable	not applicable	not applicable	not applicable	not applicable	not applicable	not applicable	not applicable	not applicable	not applicable
**Localization** **appendicular**	reference	reference	reference	reference	reference	reference	reference	reference	reference	reference	reference	reference	reference	reference	reference
**axial (head,** **neck, trunk)**	169.01	−229.74, 567.76	**0.002**	−211.10	−797.88, 375.62	0.46	−89.73	−908.39, 728.93	0.82	−119.60	−575.95, 336.65	0.59	−314.90	−1676.93, 1047.06	0.63
**Benign soft-tissue** **tumor (n = 213)**															
**Age**	16.29	0.42, 32.15	0.04	0.756	−6.20, 7.715	0.83	5.24	−2.40, 12.87	0.18	0.2511	−0.84, 1.34	0.65	26.53	8.84, 44.22	0.004
**Gender** **male**	reference	reference	reference	reference	reference	reference	reference	reference	reference	reference	reference	reference	reference	reference	reference
**female**	280.60	−243.33, 804.44	0.29	46.05	−172.08, 264.18	0.68	114.86	−124.11, 353.82	0.34	−23.45	−59.68, 12.79	0.20	515.70	−100.63, 1132.03	0.10
**Grade**	not applicable	not applicable	not applicable	not applicable	not applicable	not applicable	not applicable	not applicable	not applicable	not applicable	not applicable	not applicable	not applicable	not applicable	not applicable
**Localization** **appendicular**	reference	reference	reference	reference	reference	reference	reference	reference	reference	reference	reference	reference	reference	reference	reference
**axial (head, neck, trunk)**	21.60	−594.30, 637.50	0.95	−17.69	−268.97, 233.60	0.89	234.58	−42.02, 511.18	0.10	5.99	−37.39, 49.38	0.79	180.70	−569.78, 931.19	0.64
**Benign deep soft-tissue** **tumor (n = 172)**															
**Age**	13.01	−2.12, 28.13	0.09	2.38	−3.62, 8.37	0.43	5.54	−3.47, 14.54	0.23	0.37	−0.94, 1.67	0.58	26.59	8.57, 44.61	0.004
**Gender** **male**	reference	reference	reference	reference	reference	reference	reference	reference	reference	reference	reference	reference	reference	reference	reference
**female**	−1.82	−524.09, 520.46	0.995	144.49	−44.13, 333.12	0.13	127.00	−157.82, 411.82	0.38	−30.23	−75.17, 14.71	0.19	382.50	−285.27, 1050.29	0.26
**Grade**	not applicable	not applicable	not applicable	not applicable	not applicable	not applicable	not applicable	not applicable	not applicable	not applicable	not applicable	not applicable	not applicable	not applicable	not applicable
**Localizationappendicular**	reference	reference	reference	reference	reference	reference	reference	reference	reference	reference	reference	reference	reference	reference	reference
**axial (head,** **neck, trunk)**	59.61	−569.86, 689.07	0.85	48.07	−174.99, 271.13	0.67	291.60	−39.66, 622.89	0.08	8.75	−45.72, 63.22	0.75	422.40	−398.51, 1243.35	0.31
**Benign superficial soft-tissue** **tumor (n = 41)**															
**Age**	43.61	−17.13, 104.36	0.15	−5.39	−36.71, 25.93	0.72	−0.16	−0.97, 0.64	0.67	−0.29	−1.11, 0.52	0.47	30.60	−23.82, 85.01	0.26
**Gender** **male**	reference	reference	reference	reference	reference	reference	reference	reference	reference	reference	reference	reference	reference	reference	reference
**female**	1364.80	−196.65, 2926.25	0.08	−316.80	−1170.00, 536.33	0.44	6.39	−15.79, 28.57	0.55	6.41	−16.34, 29.15	0.57	1137.60	−350.82, 2625.95	0.13
**Grade**	not applicable	not applicable	not applicable	not applicable	not applicable	not applicable	not applicable	not applicable	not applicable	not applicable	not applicable	not applicable	not applicable	not applicable	not applicable
**Localization** **appendicular**	reference	reference	reference	reference	reference	reference	reference	reference	reference	reference	reference	reference	reference	reference	reference
**axial (head,** **neck, trunk)**	−302.60	−2074.28, 1469.01	0.73	−263.20	−1218.54, 692.14	0.56	−9.32	−33.62, 14.99	0.42	−6.88	−32.90, 19.14	0.59	−657.90	−2405.68, 1089.96	0.45

^g^: PI, patient interval. ^h^: DI, diagnostic interval. ^i^: PCI, primary care interval. ^j^: SCI, secondary care interval. ^k^: TCI, tertiary care interval. ^l^: TI, total interval.

## Data Availability

The data presented in this study are available on request from the corresponding author.

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
