# Peer review of "Enhancing Healthcare for Sarcoma Patients: Lessons from a Diagnostic Pathway Efficiency Analysis"

_cancers, 2023, doi:10.3390/cancers15194892_

Round 1
Reviewer 1 Report
In the manuscript by Elyes et al. the authors investigated the healthcare efficacy of sarcoma diagnostic processes in the Swiss setting. They collected a unique data set and carefully analysed the clearly presented data. These sorts of studies are essential in identifying patient care-related bottlenecks that may lead to changes in policy.
The identified problems and the presented system might serve other countries (even if their system is different) to lift over elements from this study.
Except for a minor comment on the lack of punctuation after "et al" that should be "et al." no more comments are to this manuscript.
Minor edit as indicated above.
Author Response
We truly appreciate the comments by this reviewer. Yes, we have now changed “et al” to “et al.”.
Reviewer 2 Report
This is an interesting paper describing the interval between symptom onset and tissue diagnosis of sarcomas, which is an important factor of survival. The result will impact future care design and delivery.
Minor comments:
Was pediatric sarcomas also included? If they are included, can the authors also compare adult vs. pediatric sarcomas in tables and discuss their findings?
Abstract - (1)(2)(3)(4) may not be necessary here.
Methods 2.2 and 2.3: What does physicians "involved" mean? Please explain.
Figure 2 is very useful. Please make the fonts bigger.
In the authors setting, should all sarcoma patients be referred to a tertiary center or can they also be treated at a secondary center? Please describe in the text. If they can be treatd in a secondary center, what was the proportion and how did the authors deal with missing data problem in tertiary care?
The last date of histologically confirmed diagnosis - Does this mean the date of biopsy/initial surgery or the date of final pathological report? Would there be another interval between biopsy and pathological report that may also be significant? (e.g. days required to obtain pathological diagnosis for a difficult sarcoma)
Line 192, with increasing age, do the authors mean the increase of every 1 year of age? Please specify.
Table 1. The comparisons are not easy to understand.
When the likelihood of A vs. B was shown, which category is the reference? Please speciy.
In the first comparison, bone vs. STS, was the three types (depth) of STS considered as a total group that was compared with bone sarcomas?
In the 2nd comparison, was all sarcomas combined together to be compared with all benign tumors? Please clarify.
Footnotes under the table may be useful.
Table 2. What comparisons do the P values represent? Please specify in the table or footnotes.
Can the authors compare the likelihood of bone vs. STS by grading?
Should the comparisons (OR, 95%CI, P values) of Region be located at the same row of the subtitle "Region"?
Table 3. The abbreviations should be spelled out in the table or in the footnotes.
If the Editor allows, please make the P values that are considered significant into BOLD font to make them more clear.
The secondary care interval (SCI) looks quite different from primary and tertiary care intervals in many rows. Please also describe in Section 3.6.1.
Lines 369-373: Do the authors know the proportion of axial sarcoma patients had CNS sarcoma?
Line 399: The concept of IPU may be supported by citing references.
Line 412, "contact": Did the authors mean context? Please confirm and explain.
Author Response
- Was pediatric sarcomas also included? If they are included, can the authors also compare adult vs. pediatric sarcomas in tables and discuss their findings?
Thanks you very much for this insightful comment. We do certainly agree that this is a very important issue we did not address. However, because there were only 38 pediatric tumor included, the numbers are too small to draw meaningful statistical conclusions. However, we did add the following sentence to the discussion section, at the end of the third paragraph, as follows: [41]. In our study, there were 38 pediatric tumors (patient age 2-18 years), 31 bone tumors (20 BS, 11 benign bone tumors), 7 soft-tissue tumors (2 STS; each one superficial and deep; 5 benign soft-tissue tumors, all with deep location). The numbers were too low to compare pediatric with adult tumors.
- Abstract - (1)(2)(3)(4) may not be necessary here.
Thank you very much for pointing this out. We adapted accordingly as per suggestions and have removed the numbering from the abstract.
- Methods 2.2 and 2.3: What does physicians "involved" mean? Please explain.
We absolutely agree with this reviewer that our writing style was indeed ambiguous. For this reason, we adapted these expressions in the text as follows:
This was done because it was not possible to distinguish between 1) the absence of physician-directed care and 2) no documentation of a physician visit in the primary care interval. Named patients were listed as not available (NA) in Figure 3 and 4 under primary care interval. The same reasoning was used for missing data on the secondary care interval. These patients were also listed as NA in Figure 3 and 4 under secondary care interval. If it was clear from the medical records that a primary or secondary care physician was not involved (e.g., because it was an incidental finding in the context of other examinations in the secondary care interval or because the referral letter from the general practitioner described it as such), the patients were listed in Figure 3 and 4 under “Absence of physician-directed care”.
- Figure 2 is very useful. Please make the fonts bigger.
Thank you a lot for this helpful comment. We enlarged the figure as requested. We certainly hope that this will satisfy this reviewer’s expectation.
- In the authors setting, should all sarcoma patients be referred to a tertiary center or can they also be treated at a secondary center? Please describe in the text. If they can be treated in a secondary center, what was the proportion and how did the authors deal with missing data problem in tertiary care?
This is obviously the multi-million dollar question, which is extremely important in light of our findings. We do thank this reviewer to teasing this out. We have mentioned the following text in the introduction section to point to this issue:
“To counteract the complex nature of mesenchymal tumors, which leads to diagnostic challenges and suboptimal treatment courses and outcomes, centralization or regionalization of diagnosis and treatment of sarcoma patients is advocated [9-12]. However, the feasibility of such centralization or regionalization depends on the availability of the necessary capacity, including the presence of sarcoma specialists. Otherwise, there could be a backlog of patients in the tertiary care sector if referrals from the secondary care sector exceed capacity.“
Based on this reviewer’s comment, however, we also believe that we have to draw the attention to the clinical implications of our findings. For this reason, we rewrote the conclusion part to address and reflect this reviewer’s suggestions. We certainly believe that this greatly enhances the message of this manuscript. The conclusion now reads as follows:
Switzerland's efficient healthcare system, cost does not guarantee expedited sarcoma diagnosis, possibly due to its multidisciplinary nature. Key factors such as older age, larger tumor size, and axial localization are associated with higher malignancy risk, underscoring the need for shorter diagnostic intervals. Further research is essential for guiding clinicians in sarcoma suspicion. To improve patient outcomes through reduced total and diagnostic intervals, focus must be placed on shortening the patient and secondary care intervals. This necessitates targeted patient education and specialized physician training. In light of our findings, we advocate for the regionalization or centralization of sarcoma care. While secondary care institutions need not be categorically excluded from sarcoma management, their involvement should be contingent upon active collaboration with a multidisciplinary team or sarcoma board from a tertiary care institution, particularly when complex treatments are required. Given these considerations, the logical next advancement for a sarcoma center is the establishment of Integrated Practice Units (IPUs), in alignment with Value-Based Health Care (VBHC) principles. IPUs offer the added benefits of transparently assessing and sharing treatment metrics and quality indicators within a collaborative network.
The second part of the question of this reviewer refers to the missing data: As we defined the endpoint of the total interval to be the tertiary care center/ sarcoma center, we did not include patients that were never in contact with a sarcoma center but only those presented to the SSN-SB. Therefore, no missing data was not encountered. We did mention this in the M&M section, as follows:
Cp Material & Methods 2.1 or discussion: “In addition, a selection bias was found for those patients who presented at a sarcoma center. That is, someone thought of the possibility of a sarcoma diagnosis during the diagnostic interval and involved a sarcoma center. Patients for whom this possibility was not considered may never have been diagnosed with a mesenchymal tumor, thereby remaining within the diagnostic interval indefinitely.”)
- The last date of histologically confirmed diagnosis - Does this mean the date of biopsy/initial surgery or the date of final pathological report? Would there be another interval between biopsy and pathological report that may also be significant? (e.g. days required to obtain pathological diagnosis for a difficult sarcoma)
Exactly, we absolutely see this interesting point: “date of histologically confirmed diagnosis” means the date of final pathological report. Herein, we adapted/used the definition of the total interval as suggested by Soomers et al. to strengthen the comparability of data, and they did not include another interval we didn’t do that either. However, the idea you mention is certainly interesting. In fact, we indeed addressed this very issue in the past (Wellauer H et al, Sarcoma Volume 2022, Article ID 7949549, 6 pages) and found great variances, from institution to institution, heavily influenced by the fact whether the analysis was directly performed by a sarcoma reference review or not, at least in our set-up.
- Line 192, with increasing age, do the authors mean the increase of every 1 year of age? Please specify.
Thank you very much for this clarifying comment. Yes, you are correct. We have therefore rephrased to: “With a 1 year increase in age,…”.
- Table 1. The comparisons are not easy to understand.
- When the likelihood of A vs. B was shown, which category is the reference? Please specify.
We’d like to thank this reviewer for this clarifying comment. STS was used as a reference, and we added this information now at the bottom of the table to clarify.
- In the first comparison, bone vs. STS, was the three types (depth) of STS considered as a total group that was compared with bone sarcomas?
This is absolute correct. We tried to clarify this point now by using soft-tissue sarcoma without specification (deep, superficial) in the “likelihood” column. We added a comment in the footnote of the table to clarify.
- In the 2nd comparison, was all sarcomas combined together to be compared with all benign tumors? Please clarify.
We absolutely agree with this reviewer, this is correct. STS and BS were compared to benign soft-tissue and bone tumors.
- Footnotes under the table may be useful.
We’d like to thank this reviewer for this helpful comment. As made urgent from the comments above, we added this clarifiying footnotes explaining what was compared to what.
- Table 2.
- What comparisons do the P values represent? Please specify in the table or footnotes.
We’d like to thank this reviewer for this clarifying comments. We added the footnote as per suggestion.
- Can the authors compare the likelihood of bone vs. STS by grading?
We’d like to thank this reviewer for this comment. As per suggestion, we performed this analysis, please see the following screenshot. We’d prefer not to add this information to the already complex manuscript since we believe, there is only prohibited clinical relevance in our context because diagnosis usually needs to be established before grading can take place.
- Should the comparisons (OR, 95%CI, P values) of Region be located at the same row of the subtitle "Region"?
Does this reviewer refer to table 1 because in Table2 there is no 95%CI etc? If so, we indeed also believe that it may be better and easier to understand when the stats are aligned on the same row as “region”.
As “appendicular” was the reference for those analyses we located the comparisons in the chosen row.
- Table 3.
- The abbreviations should be spelled out in the table or in the footnotes.
Thank you for this comment. We have added the terms as per request.
- If the Editor allows, please make the P values that are considered significant into BOLD font to make them more clear.
We have adjusted this in all tables according to your suggestion.
- The secondary care interval (SCI) looks quite different from primary and tertiary care intervals in many rows. Please also describe in Section 3.6.1.
We likely may not perfectly understand this comment. Tis reviewer is referring to the SCI versus PCI/TCI with respect to the subchapter 3.6.1. However, this subchapter describes the entire length or the total interval of the diagnostic pathway whereas we referred to SCI in subchapter 3.4. May there by a confusion? We are gladly to adapt if we do not understand correctly.
- Lines 369-373: Do the authors know the proportion of axial sarcoma patients had CNS sarcoma?
There were 55 out of 243 patients with an axially located mesenchymal tumor/sarcoma. Of these, tumors were located in the bony spine, in the skull, in the deep soft tissue of the spine, as well as superficially/epifascially from the head-neck and back regions. None was found in the central nervous system. These being very rare, we have seen them during the course of metastatic development, therefore these can not be found here.
- Line 399: The concept of IPU may be supported by citing references.
We are gladly to add morer references to this important concept. In fact, we believe that it should be used as a term to indicate the next level of development from the term sarcoma center. IPU represent the cornerstone of VBHC, and therefore we added Porter’s reference. Because our group also elaborated on this concept, we add as requested all these new references to the text in the discussion section.
-Integrated Practice Units: A Playbook for Health Care Leaders; Michael E. Porter, PhD, MBA, Thomas H. Lee, MD, MSc; Vol. 2 No. 1 | January 2021; DOI: 10.1056/CAT.20.0237
- Development of a value-based healthcare delivery model for sarcoma patients.
Fuchs B, Studer G, Bode B, Wellauer H, Frei A, Theus C, Schüpfer G, Plock J, Windegger H, Breitenstein S; SwissSarcomaNetwork.Swiss Med Wkly. 2021 Dec 24;151:w30047. doi: 10.4414/smw.2021.w30047. eCollection 2021 Dec 20.
- Line 412, "contact": Did the authors mean context? Please confirm and explain.
No, in fact we die want to refer to the word "contact" of patients with a sarcoma center. This is to express that the patients either presented on site or their situation was assessed in terms of a consultation.